# Impact of Socioeconomic Environment on Home Social Care Service Demand and Dependent Users

**DOI:** 10.3390/ijerph19042053

**Published:** 2022-02-12

**Authors:** Daniel Badell, Jesica de Armas, Albert Julià

**Affiliations:** 1Department of Information and Communication Technologies, Universitat Pompeu Fabra, 08002 Barcelona, Spain; 2Municipal Institute of Social Services, 08009 Barcelona, Spain; albert.julia.cano@ub.edu; 3Department of Economics and Business, Universitat Pompeu Fabra, 08002 Barcelona, Spain; jesica.dearmas@upf.edu; 4Department of Sociology, Universitat de Barcelona, 08007 Barcelona, Spain

**Keywords:** social support, home care, caregivers, socioeconomic features, descriptive analytics, hierarchical clustering

## Abstract

An aging population and rising life expectancy lead to an increased demand for social services to care for dependent users, among other factors. In Barcelona, home social care (HSC) services are a key agent in meeting this demand. However, demand is not evenly distributed among neighborhoods, and we hypothesized that this can be explained by the user’s social environment. In this work, we describe the user’s environment at a macroscopic level by the socioeconomic features of the neighborhood. This research aimed to gain a deeper understanding of the dependent user’s socioeconomic environment and service needs. We applied descriptive analytics techniques to explore possible patterns linking HSC demand and other features. These methods include principal components analysis (PCA) and hierarchical clustering. The main analysis was made from the obtained boxplots, after these techniques were applied. We found that economic and disability factors, through users’ mean net rent and degree of disability features, are related to the demand for home social care services. This relation is even clearer for the home-based social care services. These findings can be useful to distribute the services among areas by considering more features than the volume of users/population. Moreover, it can become helpful in future steps to develop a management tool to optimize HSC scheduling and staff assignment to improve the cost and quality of service. For future research, we believe that additional and more precise characteristics could provide deeper insights into HSC service demand.

## 1. Introduction

The growth in the number of people who are socially dependent in recent years can be partly explained by population aging and rising inequality [1,2,3], factors that are expected to continue to increase in the near future. According to the Eurostat forecasts for 2050 [4], most EU Member States will have fewer than two people of working age for every person over sixty-five. Given the severity of this problem, studies on how social services agents can help to address this challenge have attracted a great deal of attention in recent years, during which time home social care (HSC) services have grown in relevance as personalized social services [5]. This increase in social dependence can be explained by the aging of the population, but also by family structures that lead to more elderly people living alone [6,7]. HSC as a public service can provide solutions to deal with inequalities and improve the quality of life of the dependent population [8,9].

In addition to its positive social impact, some studies point out that it is more cost effective than other care models such as nursing homes [10]. Other services such as telehealth can also sustainably improve users’ wellbeing [11], but they can mostly be applied in cases with a low degree of dependency or strong care networks provided by familiars. These services are much more personalized, and users’ network and socioeconomic circumstances are key factors to determine how much demand is required [12,13]. Therefore, considering this in HSC management decisions may improve service impact on social welfare [14]. For this reason, this research proposes to study how socioeconomic factors and service demand are related.

Data analysis on health and social care systems has proven to be a useful tool for assessment and gaining new insights into administrative decision-making [15,16,17]. In this branch of study, descriptive methods can help to identify population sets (i.e., elderly or disabled population) and determine which social characteristics influence their need for public services such as home-based social care [18]. This study focused on the city of Barcelona and aimed to examine how socioeconomic features are related to the demand for services by means of descriptive analysis at the level of the area of coverage of social service centers (CSSs). Having a better understanding of how people are distributed in the territory according to profiles and needs is an essential tool for developing efficient care strategies [14]. Many works have focused on home health care [19] or other types of social care [11]. However, if they study a similar service, there are still many differences between each case [20] caused by social, cultural, and economic differences. Our research focused on the Barcelona case, which registers accessible data from HSC users. Moreover, we applied descriptive techniques such as hierarchical clustering [21] and took an original approach to set the populations. As far as we know, these methodologies have not been applied to study HSC demand and users before.

The aim of this study was to gain a deeper understanding of the dependent users’ socioeconomic environment and service needs by the application of descriptive analytical techniques. This research provides insights into the features and service needs that *Serveis d’Atenció Domiciliària* (SAD) users may have depending on their circumstances, and thus may contribute towards the distribution of SAD services in the city. This research is a first step for the optimization of scheduling and staff assignment of the public home social care services taking many considerations into account: minimization of time–distance cost of caregiver routes, balance workload, and maximization of the quality of service [22,23,24,25,26,27].

This paper is structured as follows: The next Section 1.1 formally describes the problem and frames it within the specific context of Barcelona. This city has a high percentage of elderly and dependents seeking municipal home care, and it has been growing in the last few years (according to the statistics of the Barcelona City Council, the number of home attention services users increased 20% between 2015 and 2019) [28]. Section 2 presents the solution approach along with the descriptive analytical techniques used in our methodology. This study applied principal component analysis to reduce and summarize the information. For the clustering process, hierarchical clustering methods can be useful to agglomerate similar neighborhoods, as we have seen in similar studies [29] and in other research areas such as genetics [30,31]. We define three levels of population—general population, *Serveis Socials per l’Atenció a la dependència* (SADEP) users, and SAD users (described later)—and applied these analysis techniques to explore possible patterns among their features and respective service volume outcomes.

The next Section 3 outlines the most relevant results obtained. Additional information, such as a detailed variable table, more graphical maps of Barcelona, and additional figures, are attached in the Appendix A. Section 4 presents the discussion based on these results, splitting the discussion between SADEP and SAD clustering analysis. Finally, Section 5 lists the main conclusions by linking the key facts exposed in the previous section and points out their potential impact on forthcoming research.

### 1.1. Problem Context and Description

In the city of Barcelona, HSC is in charge of delivering a set of organized actions at the user’s home. These actions consist of offering care attention and home support to users and their familiar environment with possible integration problems or personal autonomy deficiencies. This service is named *Serveis d’Atenció Domiciliària* (SAD), and it is organized by *Institut Municipal dels Serveis Socials* (IMSS) and provided by public contractors. SAD has the huge challenge of providing social home care services in a dense city where more than 20% of the population is over 65 years old, and the IMSS provides care for nearly 24,000 users. Given this volume, proper planning, forecasting, and allocation of services are crucial to increase the impact on social welfare. Many aspects are considered throughout the service allocation process, mostly in relation to the users’ socioeconomic circumstances and family environment. These two elements are related given that an adequate family or neighborhood environment can supply some of the user’s needs and an increase in the user’s dependency can deteriorate this closer setting [32]. Studies have shown that neighborhood conditions are related to the health [33] and well being [34] of its inhabitants, but this also applies to their use of public services. The health and social disparities between the neighborhoods of Barcelona [35] make the allocation of services and strategic planning even more complex. For this reason, our research focuses on the characteristics of Barcelona’s neighborhoods while also considering the overall socioeconomic features of the entire population, not just the users of home social care services.

The home care social services that Barcelona City Council provides through SAD consist of a set of coordinated actions in the user’s home. The purpose of these actions is to provide personal care, home care, and social support to users and their families with difficulties in personal autonomy and social integration. Given the large volume of service, Barcelona is divided into four zones, each served by one company, and further divided into districts. Zone 1 comprises the districts of Ciutat Vella, Sants-Montjuïc, Les Corts, and Sarrià-Sant Gervasi. Zone 2 covers the districts of Eixample and Gràcia. Zone 3 comprises the districts of Horta-Guinardo and Nou Barris. Zone 4 covers the districts of Sant Martí and Sant Andreu.

SAD is organized into social services centers (CSSs) and their areas of coverage, which are contained within a single district. Contracted companies are responsible for hiring sufficient staff to provide this service. There are four main worker profiles: technical coordinator (social worker and service planner), management coordinator (chief of staff), personal caregiver (PC, main service provider), and home care assistant (HH, home cleaner). PCs are responsible for providing the main services, which can be listed as follows:Assistance: personal care and hygiene (partial or total), food (cooking and organization), movement within the home, medication control, communication activities, and accompaniment outside the home;Educational and social: social care for underage and elderly users or users with other recognized dependencies, follow-up for adequate nutrition, and support and motivation in learning processes to reduce the impact of dependencies on the users’ basic needs;Prevention and detection: These tasks are complementary to the others and are especially important in complex services.

SAD users receive at least one service from a PC every week. They may also receive services from an HH, if scheduled. The main task of the HHs is to maintain an adequate level of hygiene in the user’s home. These tasks include only the basics, such as making the bed, sweeping and mopping the floor or cleaning the kitchen and bathroom, and an observation task to keep the coordinator informed if the user’s condition changes.

SAD service volumes, measured by service time in minutes, vary among the different CSSs, and this cannot be explained by total population volume alone. A similar variation occurs in terms of the mean time of service. This study explored which sociodemographic features may influence this phenomenon. Previous research has pointed out features that have an impact on the demand for social services [36,37,38,39]. Therefore, we analyzed demographic and economic data divided geographically by CSS coverage area.

The vast majority of SAD users have recognized dependencies in their basic activities that are registered by a social worker, in addition to their physical and mental health, home conditions, and required social services. This register is used to determine the tasks of the SAD services and estimate their weekly time allocation. All these users are listed in a system called *Serveis Socials per l’Atenció a la dependència* (SADEP), which includes all people with some form of dependency and who receive some type of social service, such as SAD, telecare, or economic support (Figure 1).

The main goal of this research is to provide a better understanding of the SAD services and its users and, if possible, to gain some insights that could help anticipate service requirements (i.e., better matching between dependents and caregivers, optimize care attendance, etc.). Therefore, this study focuses on the possible correlations between the general population and SADEP users and between SADEP and SAD users with regard to socioeconomic measures and the dimension of services.

According to previous studies, in urban contexts, there are social inequalities and territorial segregations [40] that may influence home care demands [41]. In some contexts, the socioeconomic characteristics of the population are associated with the provision of care services [36,38]. A recent study highlighted that care strategies for older dependents are determined by individuals, network characteristics, as well as contextual factors [35]. The impact of the COVID-19 pandemic on all social care services is noteworthy, especially on HSC. The impact on a service that must be delivered physically and involves many people getting in touch with many people increases the risks for caregivers and users [42]. Nevertheless, these circumstances also point out the importance of promoting long-term care services to improve social welfare [43]. This realization also increases the interest in the research of related problems, such as the adequate distribution of personal protection equipment to the required staff [44]. According to this literature, we expect that differences between territories in the urban context of Barcelona should be associated with socioeconomic and demographic characteristics of individuals and the social composition of each territory analyzed. We expect this association both in SADEP and SAD users.

## 2. Materials and Methods

### 2.1. Datasets

To carry out this research, we studied and observed correlations between features from different databases and samples. Since CSS coverage areas are the minimum geographical division of SAD/SADEP, we worked on this dimension. We focused on three different samples per CSS: overall population, SADEP users, and SAD users. As can be deduced in the previous figure, each sample is a subset of the previous one, respectively. For each sample, we aimed to study sociodemographic metrics and the amount of care received from social services (users, service time, etc.). The comparison of these samples can lead us to an in-depth characterization of SAD users, their environment, and the types of services required.

Three different data sources were used to carry out this analysis (see Table 1). We obtained sociodemographic information for each CSS zone in Barcelona from *OpenDataBCN* (ODB) [6]. Features such as the volume of people with disabilities or living alone came from this database, which was last updated between 2018 and 2020. More precise demographic and economic data were obtained from the *Instituto Nacional de Estadística* (INE) [7], where we were able to obtain data updated between 2018 and 2020. Finally, data on the management of SAD and SADEP services were obtained from Barcelona City Council. The first two databases are completely public, but the last one contains users’ private information, so it cannot be published. Most of the data can be aggregated and reduced to SAD areas (census data, geolocated users). However, some data from ODB are at the neighborhood level of detail. To address these incompatibilities, we transformed the data to CSS areas assuming an equidistribution of metrics by population volume.

### 2.2. Descriptive Analytics

In this study, we applied descriptive analytical methods to gain a deeper understanding of the data and obtain more information on user and CSS profiling. Considering the number of population metrics and the reasonable correlation between most of the variables (greater than 0.3), we decided to apply principal component analysis (PCA) to observe trends in the multivariate data. PCA is an unsupervised method for reducing dimensionality by building relevant features through linear combinations of the original variables, all of which are quantitative. The next step of our method consisted of clustering combined variables using the hierarchical method. Cluster analysis is a technique that groups individuals in a population by similarity according to their characteristics. Specifically, hierarchical clustering is a method for applying clustering that builds a hierarchical tree showing individuals as leaves and groups the branches by proximity according to the variables. Once the hierarchical tree is established, we can choose the height at which we cut the tree and determine the number of clusters.

The multivariate analysis algorithm that combines PCA and hierarchical clustering is known as hierarchical clustering on principal components (HCPC). To build a hierarchical tree in the HCPC method, we applied Ward’s criterion, which consists of linking individuals with the lowest variance error to obtain clusters with minimum internal variance. To study the optimal number of clusters, we can apply multiple statistical techniques. We chose the Elbow method since it follows the same criterion as Ward, i.e., the minimization of within-cluster variance. Once the initial cluster partition was established, we applied K-means clustering to improve this partition following the same criteria as Ward’s and Elbow’s methods.

We decided to carry out this research with a two-step methodological procedure: first, by analyzing relationships between the general population and SADEP users and, second, by comparing correlations between the socioeconomic features of SADEP users and SAD delivery. In the following section, we present the results obtained in this process.

## 3. Results

### 3.1. Preliminary Study

There are vast differences in the population density between the CSS zones in Barcelona. Given the distributions of the population and services among CSS zones, we can observe huge disparities in the volume of services in CSS zones that are clearly not related to the population volume (Figure A1 and Figure A2). In addition, we note that the average service hours (per PC or HH) per user vary significantly between CSS zones (Figure A3 and Figure A4). These differences could be related to socioeconomic factors, as pointed out by previous research [36,38].

As a preliminary study, we examined the correlation between each pair of parameters included in this research. The results obtained can be seen in Figure 2. We can observe strong correlations between demographic parameters such as percentage of females and population over 80 years old, but also between SADEP and SAD percentage of users (0.82%). Despite this, we can conclude that there are no trivial linear correlations between the SAD or SADEP metrics and the sociodemographic features of all populations.

In the next step and prior to the PCA analysis, we assessed how the economic features of the CSS can influence the volume of social services. This preliminary study allowed us to observe whether there was any evident correlation between the parameters of interest. We split the CSS areas into two groups based on mean net rent per person (MNRP), setting the dividing line at EUR 14,686, the median of this set. We then established Group 1 as the set of CSS zones below the median and Group 2 above. In Figure 3a,b, we plot the percentage of CSS users over CSS users and caregiver care time per user on the y-axis, respectively. In the first figure, we observe that SADEP users tend to also require SAD in the poorest neighborhoods. However, in the second figure, we note that social caregivers provide slightly more service time per user in wealthier areas. This phenomenon can be explained by the different profile of users depending on the economic level of the neighborhood. In the wealthier areas, users who request SAD may have more needs due to a severe level of dependence or other factors. Given these results, it seems that the volume of dependence services and economic level may be related. Our first hypothesis, based on these preliminary results, is that poorer neighborhoods have relatively more SAD users, but the volume of service can vary over a wider range of hours. In contrast, in wealthier neighborhoods, there may be fewer SAD users, but they require more service time. These phenomena could be explained by the use of private assistance by wealthier users with a low level of dependence circumstances.

### 3.2. Population—SADEP Analysis

At this point, we applied PCA to the socioeconomic data for the entire population. The fields used in this PCA are in relation to age, gender, population density, disability and MNRP. In this case, all data had to be transformed to avoid dummy correlations between variables such as population and volume of SAD users. We normalized all variables to avoid correlations arising from the volume of people in a SAD coverage area. To do this, we computed the percentages of the overall population or other subsets instead of applying quantitative measures. The PCA results after this linear transformation showed that three dimensions were needed to sufficiently describe the sample with a total of 80.41% of dataset inertia, respectively (Figure 4a). In Figure 4b, we note that the variables are diverse given that most of them are correlated with other characteristics, but are completely the opposite in other parameters. An example of this diversity is the percentage of people with third-degree disability. This shares a strong correlation with MNRP, but is entirely unrelated to the percentage of people with first-degree disability. However, we must bear in mind that three components are required to have sufficient information, while these two-dimensional representations lack the third component.

From the above results, we applied a hierarchical clustering method to group the CSS areas by PCA dimensions. The hierarchical tree was built on Ward’s criterion and cut at the expected height. This process led to the sample being split into four clusters, as seen in Figure 5. Additionally, Figure A5 and Figure A6 show how the clusters are distributed in a 3D space relative to the principal components.

These clusters are characterized by the distribution of the parameters shown in Figure 6. The first point to consider in this table is that all *p*-values are less than 0.05, so this clustering process proves to be more consistent [45]. As we can see, the first cluster is characterized by having a lower MNRP and a much smaller population volume with respect to the rest of the sample. Note that the percentage of elderly people is far below the mean, while the percentages of people over 65 with recognized disability or living alone are above the mean. In addition, this cluster has an above-average percentage of second-degree disabled people, but works in the opposite direction with the percentage of people with first-degree disability. On the other hand, the second cluster is characterized by a low MNRP and population. The distribution of this cluster is characterized by a higher percentage of disabled people, especially of first-degree severity. The CSS areas of the third cluster are distinguished by a higher population and density, as well as an aging population. Finally, the last cluster stands out for its higher MNRP, low population density, and lower percentage of disabled people over population. However, the percentage of people with third-degree disability is higher.

After applying the HCPC method, we analyzed how clusters formed by population-wide socioeconomic data influence SADEP user parameters. In Figure 7, we can observe patterns that could establish relationships between cluster characterization and SADEP metric outputs. Note that the MNRP distribution follows a similar path to that described by the cluster characterization, and the high percentages of disabled people in the second cluster are consistent across the SADEP subset. Moreover, the fourth cluster leads in the percentage of people with third-degree dependence, as well as the percentage of people with third-degree disability considering the whole population.

### 3.3. SADEP—SAD Analysis

In the following steps, we studied how the parameters of SADEP and SAD services relate to each other and whether we can determine clear relationships between these two populations using the HCPC method. For this analysis, all socioeconomic data were restricted to SADEP users and normalized with this scale. After applying this method, an analysis between the resulting clusters and SAD metrics was carried out and compared with previous results.

We thus repeated the PC method on the SADEP user parameters. As before, we saw that to reach a percentage of approximately 80% of the variance description, three components were required. In Figure A7, we can observe the results of the principal component method. Note that Figure A8 shows negative correlations between users with different degrees of dependence and population aging. Additionally, we observed an absence of correlations between the percentage of SADEP users and their MNRP. Again, we must note that these observations were made on the projection on a plane of the three-dimensional space formed by the three principal components.

The next step was to apply hierarchical clustering based on the principal components obtained. In this case, three clusters were identified (Figure 8, Figure A9 and Figure A10). As in the previous analysis, Figure 9 shows how the clusters are characterized by SADEP features. From these observations, we can see that all *p*-values are less than 0.05 as well, which supports this analysis. The first cluster can be identified by its lower MNRP and above the mean percentage of disabled people. The second cluster is difficult to identify because of its lack of differentiation of SADEP metrics and only slightly lower percentage of SADEP users compared to the global mean. The last cluster encompasses the CSS areas with the highest MNRP, percentage of people over 65, and percentage of people with second- and third-degree dependencies.

Following the path described at the beginning, we analyzed how the clusters based on the SADEP parameters relate to the SAD metrics. To do so, we plotted several boxplots, one for each metric. In Figure 10, we can observe that SADEP users in the first cluster are more likely to require SAD services. In addition, we see that the same cluster has the highest percentages of disabled people. In the third cluster, there is a greater time spent on home cleaning services compared to the other two clusters.

## 4. Discussion

The results obtained in this study can lead to several conclusions for discussion. Firstly, comparisons between the population and SADEP sets help to understand how socioeconomic circumstances are related between people with dependencies and the overall population. Secondly, the analysis between the SADEP and SAD sets revealed the social features of people with dependencies who claim SAD and the relationships of such features with the amount of service required.

### 4.1. Population—SADEP Analysis

In the previous PCA step, we were able to reach a sufficient level of description (80.41%) with three principal components having description levels above 10%, which is a common subjective cut-off threshold [46]. This step balances the influence of several groups of variables from all populations (economic, disability, demographic) and leads us to a consistent clustering by the HCPC method. We obtained four clusters with a robust p<0.05 for at least two of the three components. Specifically, we can characterize each cluster with at least six features that have p<0.05, with disability-related fields being the most common. We then analyzed how this application of HCPCs on global population features affects SADEP features.

These results showed the consistency between the degrees of disability and dependency in Clusters 1, 2 and 4. These consistencies arise from the predicted relationship between physical or mental disabilities and self-sufficiency. While Clusters 1 and 2 are the poorest sets of CSS areas (respective MNRP means of 10,841 and 10,169), Cluster 4 is defined by the neighborhoods with the highest MNRP. Thus, it appears that, given a certain degree of disability, economic well-being does not decrease the demand for SADEP services. Cluster 1 is characterized by a lower population volume and percentage of elderly people, but a higher rate of elderly people living alone and a higher rate of elderly people with disabilities. The extent of SADEP in these neighborhoods is lower compared to the rate of users in the other clusters, and the users are mostly elderly people with lower degrees of dependency. Cluster 2 can be described by its lower population and higher rate of disabled people. As in Cluster 1, the degree of dependency and disability percentages persist, with a higher rate of disabled people and first-degree dependency. A higher percentage of users with dependency is also observed, which is explained by the lower economic levels of the neighborhoods. The neighborhoods in Cluster 4 are characterized by a lower rate of disabled people and a higher MNRP and rate of severely disabled people (with a degree of disability over 65%), which is consistent in terms of the economic level and degree of dependency of its SADEP users. As for Cluster 3, which shows a higher population density and rate of elderly people, no remarkable values of SADEP characteristics are observed. From this analysis, we can conclude that disability and economic features have the greatest influence on the demand for SADEP services. Specifically, we observe related distributions for clusters with lower MNRP and second degree of dependency. A person with this degree of dependency requires assistance to perform essential activities two to three times a day. This result can provide further insights on forecasting future demands for dependency assistance by CSS area.

### 4.2. SADEP—SAD Analysis

The second part of this research analyzed how SAD features are influenced by the socioeconomic environment of dependent users who require SADEP services. The PCA applied obtained a level of description of 77.47% considering three components with levels of description above 10% as in the previous analysis. From the hierarchical clustering process, three clusters were obtained with p<0.05 in the principal components that define each of them. We note the SAD features of each cluster and compared these results with the characteristics of each cluster. We can observe clear differences in SAD features between clusters in the following domains: percentage of SAD over SADEP users, income, disability, cleaning service time per user, and percentage of elderly users requiring HH services. From these variations, we can observe that although the number of SAD users over SADEP users requiring HH services is consistent, the HH service time per user is slightly inverse. Clusters 1 and 3 show the clearest difference, which could be explained by their MNRP values. Cluster 1 comprises poorer neighborhoods and a higher demand for SAD among SADEP users. This phenomenon may be due to the high proportion of disabled people in the same cluster. Conversely, in Cluster 3, which comprises wealthier CSS areas, we clearly see that the proportion of SADEP users requesting SAD decreases. We concluded that the time of HH services has a strong relationship with the MNRP values. While in poorer areas, there are more users, but less service time is provided per user, the opposite phenomenon happens in wealthier neighborhoods. This may be due to many factors: differences in users’ needs, in users’ expectations of the service, or in service time delivery capacity.

## 5. Conclusions

Our study showed the association between the socioeconomic characteristics of the population and the demand for municipal public home care services. This is explanatory research that should be considered as a first step for the optimization of the municipal HSC services (e.g., efficient visiting distribution, matching between dependent demands and care services, etc.). Our research also showed that territorial and socioeconomic inequalities in the same city reproduce inequalities in the amount of service provided. For example, poorer neighborhoods have more users and less service time per user than in wealthier areas.

Given the expected increase in demand for these services due to the aging population and the evolution of social and health care, this study contributes to a better understanding of how HSC must be distributed across the city in the future to serve users of some type of dependency. Our study has the potential to obtain a wider impact on strategic decisions in the management of social services. First, additional features such as the number of cohabitants per user, the Gini coefficient, or other economic metrics in addition to the MNRP could help to better characterize the clusters. In addition, they could provide new insights into which socioeconomic features may be most useful in forecasting variations in demand. Having systematized information on the characteristics of the users of municipal care services, their demands, and needs is essential to develop more efficient management strategies. Some examples considered in other contexts can be seen in [11], which concluded the promotion of telehealth services as a sustainable way to improve the well-being of a great volume of users. From our analysis, we can notice inequalities in service delivery between CSS areas. These measures can be used to improve the management of HSC services and reduce these inequalities. Many studies that focus on optimizing scheduling for all types of home care services [22,26,27,47,48,49,50] apply heuristics or metaheuristics. Our research can work as a first step to start building an algorithm to schedule services for SAD, as we have more information to simulate scenarios to validate the proposed solutions.

This study has certain limitations that should be noted. The precision of users’ economic information can be imprecise in some cases because it does not consider economic data from cohabitants (closer network). Another limitation is that geographical divisions do not consider socioeconomic features and are given by HSC service. Another way to split BCN into smaller areas could provide clearer correlations with HSC demand. Last but not least, this exploratory analysis did not provide predictors that could help forecast demand. We believe that we would need more precise data to try to make reasonable predictions.

These limitations notwithstanding, our study contributes to the knowledge of urban context inequalities in care services and the efficiency of HSC services. Our findings contribute to a major understanding of HSC efficiency and inequality that municipal social services should take into account. The distribution of resources should be based on social needs and equity for the sake of social justice and resource optimization. Despite not being investigated in this research, a further avenue to explore is the application of predictive analysis. This path would be useful in detecting potential changes in the demand for services and implementing the appropriate preventive policies. According to our findings, future research should be focused on how to reduce urban socioeconomic inequalities in HSC provision. This is a key objective in order to optimize the limited municipal resources in a context characterized by an increasing elderly population and demand for HSC.

## Figures and Tables

**Figure 1 ijerph-19-02053-f001:**
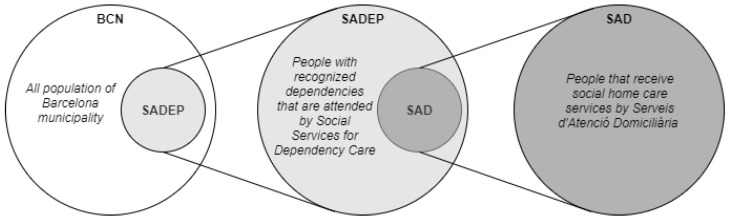
Outline of the population groups used in this research work.

**Figure 2 ijerph-19-02053-f002:**
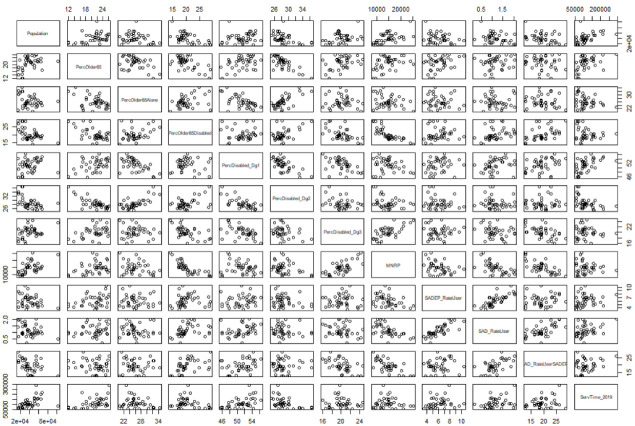
Correlation matrix of available parameters. From top-left to bottom-right: population, percentage (Perc) of population older than 65, Perc of population over 65 and living alone, Perc of population over 65 and disabled, Perc of users with a certain degree of disability, MNRP, Perc of SADEP users in the whole population, Perc of SAD users in the overall population, Perc of SAD users over SADEP users, and volume of SAD service.

**Figure 3 ijerph-19-02053-f003:**
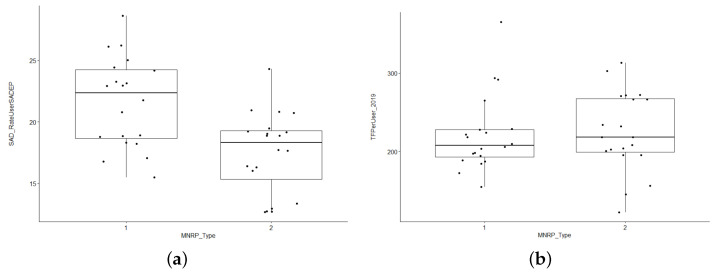
Boxplots of CSS areas grouped by MNRP classes, which are split at EUR 14,686. (**a**) Comparison of the percentage of SAD over SADEP users between MNRP classes. (**b**) Comparison of PC service time per user between MNRP classes.

**Figure 4 ijerph-19-02053-f004:**
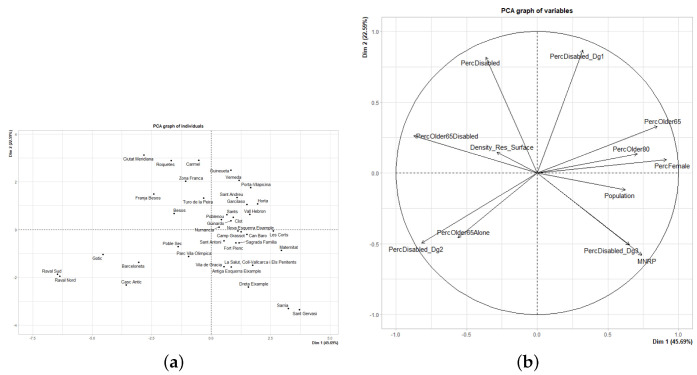
Results of PCA in the CSS areas based on socioeconomic metrics over the entire population. (**a**) CSS in the first two components. (**b**) Biplot showing the influence of parameters on the first two principal components.

**Figure 5 ijerph-19-02053-f005:**
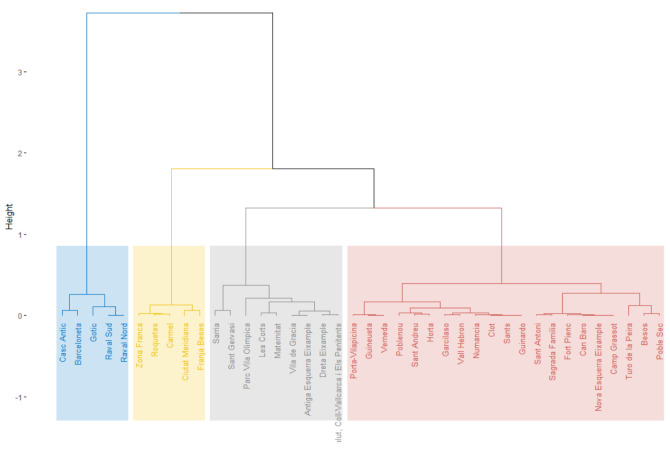
Results of hierarchical clustering principal components (HCPCs). Hierarchical tree cut to build four clusters.

**Figure 6 ijerph-19-02053-f006:**
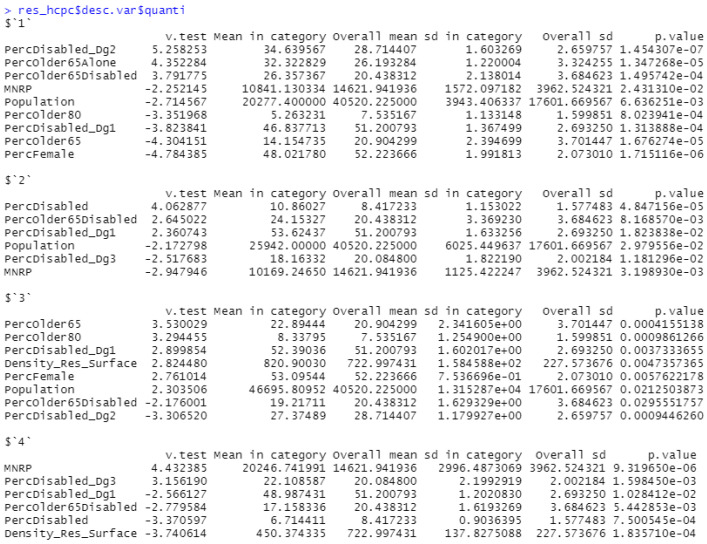
Distribution and influence of the parameters per cluster. Description of the variables specified in Table A1.

**Figure 7 ijerph-19-02053-f007:**
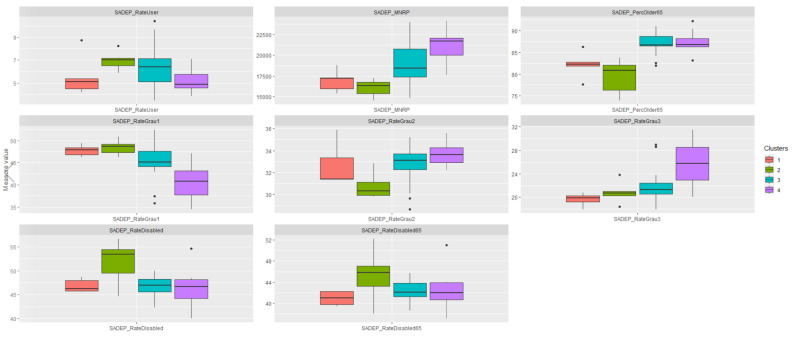
Boxplots of CSSs grouped by the HCPCs’ resulting clusters. The Y-axis shows different measures of SADEP’s features. The scale can vary between metrics. Description of the variables specified in Table A1.

**Figure 8 ijerph-19-02053-f008:**
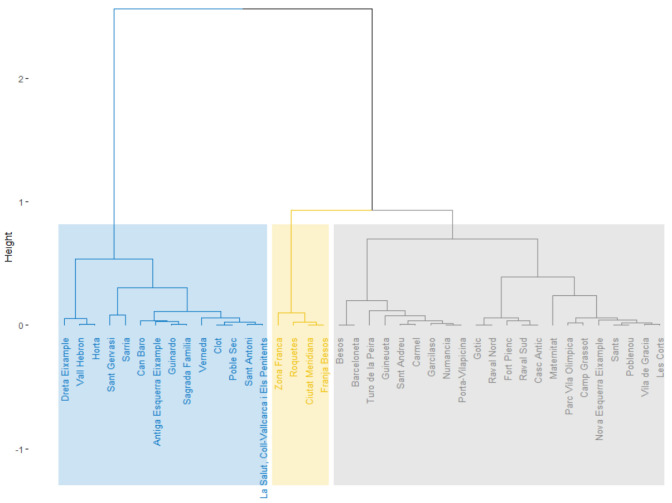
Results of HCPCs. Hierarchical tree cut to build three clusters.

**Figure 9 ijerph-19-02053-f009:**
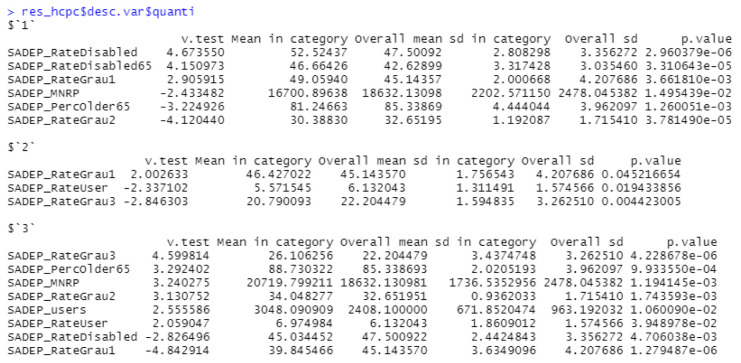
Distribution of SADEP features per cluster. Description of the variables specified in Table A1.

**Figure 10 ijerph-19-02053-f010:**
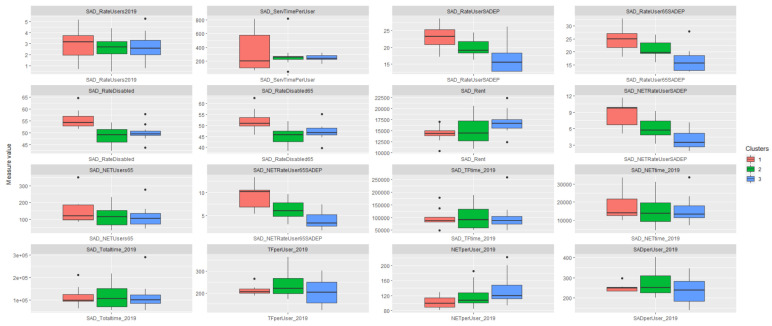
Boxplots of CSS grouped by the HCPCs’ resulting clusters of SADEP characteristics. The Y-axis represents different SAD features. The scale can vary between metrics. Description of the variables specified in Table A1.

**Table 1 ijerph-19-02053-t001:** Table describing all the data used in this study. Columns include, from left to right, data sources, type of privacy (availability), last update (given the variety of tables, years may differ in the same data source), and features included.

Data Source	Last Update	Features
INE (Public)	2018–2020	Population/age/gender/economic
OpenDataBCN (Public)	2018–2020	Live alone/disability
BCN Council (Private)	2019–2020	Volume and type of service/SADEP and SAD
		Users/dependency/economic/disability

## Data Availability

Data coming from OpenDataBCN [6] and INE [7] sources are public. However, data from SADEP services contain sensitive information on users and cannot be published publicly.

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
