# Peer review of "Impact of Socioeconomic Environment on Home Social Care Service Demand and Dependent Users"

_ijerph, 2022, doi:10.3390/ijerph19042053_

Round 1

Reviewer 1 Report

I am pleased to have the opportunity to review this research paper. This study attempted to explore the impact of Socioeconomic Environment on Home Social Care Service Demand and Dependant Users. Although the topic of this research study is interesting and fits within the journal scope, I think authors should apply the comments indicated below to increase the quality of research justification, contributions, and findings. The manuscript know lacks in scientific style and structure.

  1. First of all, paper research gap. Please improve this part in the introduction section. The introduction is very general and lacked alignment to the research findings, no discussion was provided to derive the implication from. Theoretical and pragmatics implications are vague and need to be better aligned with this paper's theoretical underpinnings and proposed process. Furthermore, there is insufficient support and weak arguments in support of the objective that is proposed as well as the model developed. In the final part of the introduction, the objectives proposed originality and gap that would be better covered. Also how the author will perform the methodology. I strongly suggest considering: “Optimising the routing of home health caregivers: can a hybrid ant colony metaheuristic provide a solution?. British Journal of Healthcare Management, 26(7), 192-196.”
  2. Explain and add exactly the methods presented in the literature were reduce costs, save time and improve home healthcare (we refer the authors to consider the papers:
  • General variable neighborhood search for home healthcare routing and scheduling problem with time windows and synchronized visits. Electronic notes in discrete mathematics, 58, 63-70, 2017.
  • Optimising the travel of home health carers using a hybrid ant colony algorithm. In Proceedings of the Institution of Civil Engineers-Transport(pp. 1-12). Thomas Telford Ltd, 2021.
  1. In the arrangement of the structure of the article, the key part should be highlighted, and the key part needs to be described in detail.
  2. The authors must follow the manuscript format of the journal, special in the section of references.
  3. The literature review is too simple to reach the standard of the IJERPH Literature analysis needs to be improved.
  4. What are the theoretical and management implications of this study?
  5. In conclusion, systematize the advantages and limitations of your research study. And, the conclusion needs to be enhanced further.
  6. Discussion including detailed implications should also be part of the existing discussion.

Reviewer 2 Report

Comments to Author:

This paper was written very well and has a good opportunity to be published in this great journal, after addressing the following concerns in the first round:

- I encourage you to add more detail about your core contributions in the abstract. Abstract has five-section and you should follow the best practices in your area! Please also mention the novelties in the abstract.

- The keywords are not standard.

- Long paragraphs.

- Please bring some facts and figures in the introduction to support the ideas. Especially from www.statista.com

- Literature review is very short and old! Now we are at the beginning of 2022! You have not covered the knowledge edge! Please clarify the contribution of the paper according to the research gap. Many recent papers in the area can be added to the literature review. You have not referred to the main works in this area. I do not propose you any special reference due to ethical issues. Just as the sample I suggest some works that I saw in this area in 2021 and 2022 as examples and show that you can review and cite more works in this area. Especially please mention the recent pandemic and discuss in the introduction that how it can affect your assumptions. The authors should review the recent works in SC areas. I simply search and find the following works:

Covid:

- Disaster relief supply chain design for personal protection equipment during the COVID-19 pandemic. Applied Soft Computing, 107809.

Healthcare:

Bi-level programming for home health care supply chain considering 
outsourcing. Journal of Industrial Information Integration, 100246.

Sustainability: (It is better to discuss on sustainability and environmental issues in the introduction, at least one paragraph)

Sustainable planning and decision-making model for sugarcane mills considering environmental issues. Journal of environmental management,
303, 114252.

- Please add more references from the target journal. The number of references is very short for such a paper. Please be at the knowledge edge. In the introduction, you brought more than five references from 2021 and in the LR table, you did not mention even one from 2021!!!!

- Please add a research gap section.

- Please, clarify which one of the assumptions is new in this area in the problem definition.

- Check the English presentation of this paper to remove the typos mistakes.

- Findings, limitations, and recommendations of this paper can be discussed more in the conclusion section.

- Please bring and focus on future research directions.

I go with a major revision in this step and waiting for your corrections. Then, I give you my technical comments.

Reviewer 3 Report

The paper is interesting. It gives the state of the art around the impact of socioeconomic environment on home social care service demand. 

1) I suggest adding a clearly formulated aim of the paper in the abstract

2. In the Introduction section I suggest separating the research objective from the research method. The objective says what we want to get. The research method, how we want to get it. 
For example: The aims of this research is to get a deeper understanding of the dependent user' socioeconomic environment and service requirements. 

3) In the Introduction section, I propose to add the research hypothesis(s) and then in the text refer to their confirmation or rejection. 

4) I suggest that you expand the Discussion section to show exactly what the analysis can be used for and by whom. I also suggest adding potential barriers or limitations to using the described analyses in these areas. 

5) In the Conclusions section I suggest adding in which areas these analyses can be useful for strategic decisions in social services management.

Round 2

Reviewer 1 Report

Accept after minor revision. Minor English Corrections

Author Response

Thank you for your comments. English revision was made by editing services for the most part of this work, except for the new additions of the previous review round. In this revision, we worked on improving written communication in the mentioned new content. All changes are colored in orange.

Reviewer 2 Report

The authors did have not addressed my comments precisely. They have not covered the literature reviews carefully and have not addressed the proposed areas. There is no meaningful finding. Figures are still not clear and this work can not be considered as a scientific paper.

I invite the authors to do the comments precisely and give them another chance to do them.
